# Improving an Acoustic Vehicle Detector Using an Iterative Self-Supervision Procedure

**Birdy Phathanapirom** [1,*]**, Jason Hite** [1]**, Kenneth Dayman** [1]**, David Chichester** [2] **and Jared Johnson** [1]

1 Oak Ridge National Laboratory, Oak Ridge, TN 37830, USA
2 Idaho National Laboratory, Idaho Falls, ID 83415, USA
* Correspondence: birdy@ornl.gov

**Abstract:** In many non-canonical data science scenarios, obtaining, detecting, attributing, and annotating enough high-quality training data is the primary barrier to developing highly effective models. Moreover, in many problems that are not sufficiently defined or constrained, manually developing a training dataset can often overlook interesting phenomena that should be included. To this end, we have developed and demonstrated an iterative self-supervised learning procedure, whereby models are successfully trained and applied to new data to extract new training examples that are added to the corpus of training data. Successive generations of classifiers are then trained on this augmented corpus. Using low-frequency acoustic data collected by a network of infrasound sensors deployed around the High Flux Isotope Reactor and Radiochemical Engineering Development Center at Oak Ridge National Laboratory, we test the viability of our proposed approach to develop a powerful classifier with the goal of identifying vehicles from continuously streamed data and differentiating these from other sources of noise such as tools, people, airplanes, and wind. Using a small collection of exhaustively manually labeled data, we test several implementation details of the procedure and demonstrate its success regardless of the fidelity of the initial model used to seed the iterative procedure. Finally, we demonstrate the method's ability to update a model to accommodate changes in the data-generating distribution encountered during long-term persistent data collection.

**Keywords:** data fusion; self-supervised; semi-supervised; classification; infrasound

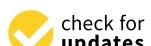



## 1. Introduction

### 1.1. Context

Monitoring activities at nuclear fuel cycle facilities such as nuclear reactors and fuel reprocessing plants is a necessary step in ensuring the peaceful use of nuclear technology around the world. Substantial resources and human capital are continuously devoted to this mission worldwide through organizations such as the International Atomic Energy Agency (IAEA). One of the primary methods employed by the IAEA is onsite, in-person inspections, which are especially resource intensive, requiring an inspector to travel to the site, visually inspect the site, make measurements, and potentially collect samples for later analysis, and interpret the data. Persistent monitoring with autonomous systems and non-invasive sensor systems could substantially reduce the effort required to verify the peaceful use of nuclear technology.

To this end, the Multi-Informatics for Nuclear Operations Scenarios (MINOS) venture is intended to provide a test bed for the development and demonstration of multimodal analytics and sensor fusion techniques applied to the monitoring of activities within a complex, multiuse nuclear facility. The test bed is located at Oak Ridge National Laboratory (ORNL) in Oak Ridge, Tennessee, and consists of two facilities: the High Flux Isotope Reactor (HFIR), an 85 MW$_{th}$ nuclear reactor designed mainly for isotope production and neutron research, and the collocated Radiochemical Engineering Development Center (REDC), which houses laboratories for the chemical processing and extraction of isotopes

produced primarily by neutron irradiation in HFIR. Characterization of activities within each facility, as well as transfers of materials and equipment between the facilities, are of interest to the venture. The site has been instrumented with networks of sensors across five modalities (gamma radiation, low-frequency acoustic, seismic, and electromagnetic and thermal imagery) in order to monitor the reactor's operation and power; to detect and characterize chemical process activities within REDC; and to detect transfers of radiological material within and around the test bed.

Our team's primary interest is to selectively detect vehicle-borne movements of radiological material, characterizing the type of material(s) in each movement, and tracking these transfers over time. Our approach is to specifically detect the large trucks used for moving heavily shielded radiological material (e.g., material targets irradiated in HFIR to produce desirable isotopes such as $^{252}$Cf that are then separated and purified in REDC), down-select detected vehicles that contain radioactive payloads, identifying the specific types of material, and then combining these detections in a temporal model. This paper will focus on the robust detection of large trucks.

### 1.2. Problem Statement, Challenges, and Solution Overview

Previous work by Hite et al. [1] constructed a binary classifier designed to detect vehicles using low-frequency (sub-400 Hz) acoustic data persistently collected using a network of smart phones running specialized software distributed around HFIR and REDC. This capability was shown to be reasonably effective; however, the performance was sub-optimal and was not sufficient to add value when fused with other modalities and associated analyses. The primary limitations in developing the classifier presented in Hite et al. [1] are the large number of person-hours required to process and characterize interesting events in the streamed acoustic data to assemble a reliable set of training data. These data processing tasks were originally performed manually, limiting the total amount of labeled training data that could be produced. This manual analysis gave a limited number of labeled instances (e.g., large trucks passing by the sensor or airplanes flying overhead), each of which are nonstationary. To yield a larger number of stationary examples, subsamples were obtained from each instance, and these samples inherit the label from the larger instance. Because acoustic phenomena change over time (e.g., a truck is approximately silent while at a stop sign but loud before and after), some samples are necessarily mislabeled.

To address these limitations and improve the vehicle detection capability, we aimed to develop and demonstrate a semi-supervised/self-supervised learning algorithm to incrementally improve the performance of the original classifier constructed in Hite et al. [1] by autonomously processing new unlabeled data and producing new training data examples. It is unclear whether it is more appropriate to call our method semi-supervised or self-supervised. While our iterative approach initially uses a small amount of labeled data and adds unlabeled data (suggesting the semi-supervised descriptor), these data are continuously diluted or outright discarded in later generations, where the method entirely relies on events and associated labels detected and attributed from the streamed data autonomously by the algorithm (suggesting the self-supervised moniker). In the remainder of this paper, we will adopt the term "self-supervision" to describe our method.

### 1.3. Related Work

Other fields such as Intrusion Detection [2], medical image classification [3,4], and Offensive and Hateful Speech monitoring and detection [5] also suffer from having limited and difficult to annotate data and have benefited from the inclusion of unlabeled data in the training of classifiers.

Van Engelen and Hoos [6] describe an example in which semi-supervised learning is used in the context of document classification (e.g., news articles). A simple supervised classifier may learn to recognize the word "neutron" as an indicator of a physics-related article; however, the selection of the word "neutron" may be unintentionally influenced by

our choice in articles to hand label as training data for the classifier. That classifier would fail if a physics-related article were presented that did not contain the predictive words used in the training set, such as an article on particle accelerators. The omission of the keyword "particle accelerator" from the training set could result from time constraints due to the need for human annotation or even simple oversight. However, if the unlabeled data are considered, there may be documents that connect the word "neutron" to the phrase "particle accelerator" via co-occurrence with "quark". That connection would guide the classifier to correctly identify those documents as pertaining to physics, despite never having seen the phrase "particle accelerator" in the labeled training data.

In this example, incorporating unlabeled data into the analysis yields a more powerful classifier by enabling the discovery of new relationships and features that were not represented in the labeled data. Cozman, Cohen, and Cirelo [7] assert that inclusion of unlabeled data can improve classifier performance so long as modeling assumptions match the model-generating data, where one common violation is the inclusion of unlabeled data from unseen classes [8]. Although initially developing a larger and more diverse corpus of training data by conventional means (e.g., including additional keywords) could achieve the same result, the increase in performance must be weighed against the additional cost of developing the larger dataset. Moreover, it is likely that a human-developed training dataset will overlook some interesting feature, example, or other source of variation. The autonomous development of training data using a self-supervised approach avoids this additional cost and may be more robust to overlooking interesting data by allowing larger volumes of data to be processed without human intervention.

### 1.4. Outline

The paper is structured as follows. Section 2 provides additional information regarding the MINOS test bed, including description of the infrasound data collection, characterization, processing, and use in training our reference classifier. Further, the iterative self-supervised learning procedure used in this work is described in full in this section. Section 3 contains a validation of said procedure, as well as results from experiments that explore several design choices in the procedure in which the original labeled data are continuously diluted or outright discarded in later generations. Finally, Section 4 summarizes our findings and highlights three primary use cases of our self-supervision approach.

## 2. Materials and Methods

The data sources and analysis methods used in this work are described in this section. The relevant infrasound data, which are collected persistently at the MINOS test bed, is described in Section 2.1. Section 2.2 describes how a vehicle signature and associated classifier are derived from a small amount of hand-labeled data that is used to represent the capability of canonical supervised learning in our limited ground truth scenario. Section 2.3 expands on the self-supervised learning paradigm given previously, with the specific aim of clarifying where in the taxonomy of unsupervised-to-supervised learning this work falls.

### 2.1. Measurements and Signature Identification

An aerial view of the MINOS test bed is shown in Figure 1. A diverse network of sensors is distributed in and around the site, including acoustic, seismic, radiation, thermal imagery, and electromagnetic field strengths. The orange line in Figure 1 marks the primary material transfer pathway of interest between HFIR and REDC, though the site is open to general vehicle and pedestrian traffic. As in Hite et al. [1], we emphasize the goal of identifying *any* vehicle, with parallel work focusing on further distinguishing vehicle type, movement patterns, and the content transported.

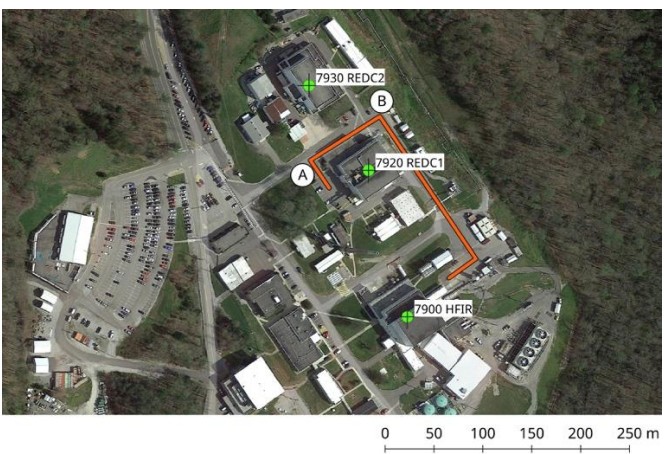

**Figure 1.** Aerial view of MINOS test bed at ORNL with main vehicle transfer route between HFIR and REDC highlighted (orange) and two acoustic sensors indicated (white circles, A and B). Satellite imagery ©2019 Google, used with permission.

Two Samsung Galaxy S10 smartphones serve as acoustic sensors, referred to as sensors A and B, and are positioned in locations suitable for tracking vehicle movements along the main transfer route between HFIR and REDC. These sensors run the RedVox software (https://www.redvoxsound.com, accessed on 1 December 2021), which records each phone's onboard sensors and reports the data to a central server. Included in the reported data are audio recordings made using each phone's internal microphone, which are sampled at a rate of 800 samples per second and are thus band-limited to a maximum frequency content of approximately 400 Hz.

Most statistical analyses and machine learning algorithms assume stationary data. However, because real vehicle movements would necessarily involve accelerating, coasting, braking, etc., we expect changes in vehicle throttle and associated engine noise to lead to nonstationary acoustic data. Moreover, the long acoustic measurements collected at the testbed exhibit characteristics that are specific to the MINOS testbed. For example, if a sensor is located near an intersection, vehicles are expected to consistently coast, brake, pause, throttle, and turn at this intersection. The resulting pattern would affect the measured acoustic information, and an analysis could be developed to recognize this characteristic sequence of events; however, such an analysis is unlikely to be more generally useful as it would likely be overfit to the particular sensor and its placement. Instead, we aim to develop a more general vehicle detection capability that relies on fundamental physical phenomena associated with any vehicle-borne material transfer. Accordingly, we have adopted a process of subsampling data to produce smaller observations of approximately 2–3 s referred to herein as "clips." The intention was to learn patterns from these clips that were generally indicative of vehicles and were invariant to sensor placement (i.e., could be used for any sensor in the testbed).

Given a clip, the short-time Fourier transform [9] was applied to the audio waveform to produce a spectrogram that was used as features for vehicle detection. Beginning from a clip with fixed sample rate $s$ and duration $t$, the short-time Fourier transform of the segment was computed using a window length $w$ (in number of samples) and window overlap fraction $f$, yielding a spectrogram matrix whose columns were then stacked to form a feature vector.

Hite et al. [1] observed a characteristic signal attributed to vehicles, such as that seen in Figure 2, wherein a distinctive pattern of fundamental peaks and their harmonics can be seen. The peak frequencies vary in time but move in sync while maintaining their spacing in frequency. A human operator was able to consistently identify this vehicle signal in a spectrogram with minimal practice. These hand-labeled data were used to train the original classifier, serving as the basis of the work described in Section 2.2. For comparison, other common events as viewed in the frequency domain are shown in Figure 3 (standard

background that appears as broadband noise with occasional narrowband peaks) and Figure 4 (wind noise). Note that several events may occur simultaneously, and a successful classifier should be capable of distinguishing between a vehicle even in the presence of unpredictable nuisance events such as wind noise and the unaccompanied nuisance event.

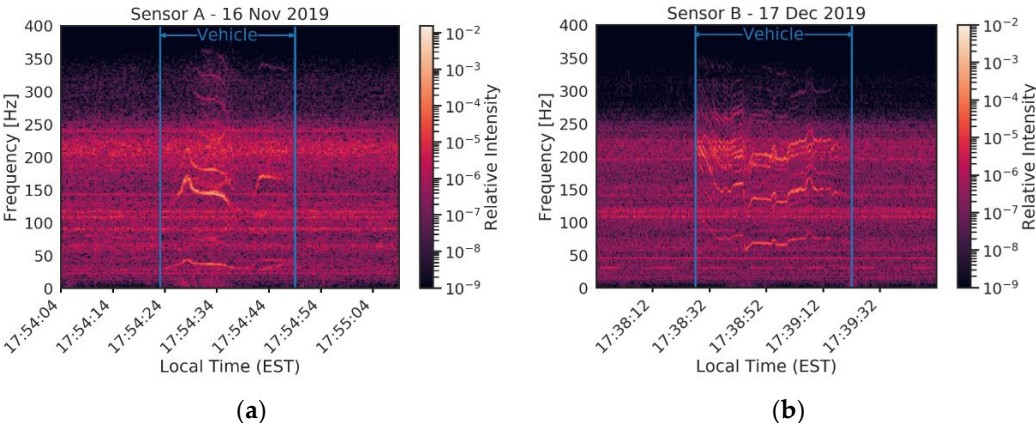

(**a**)                                                                (**b**)

**Figure 2.** Characteristic vehicle signal in frequency domain as recorded by (**a**) sensor A and (**b**) sensor B. While the characteristic vehicle signature (undulating base frequency with multiple harmonics) is visible in both measurements, note that there are considerable differences between the two signatures, making most audio analyses and simple statistical analyses inappropriate.

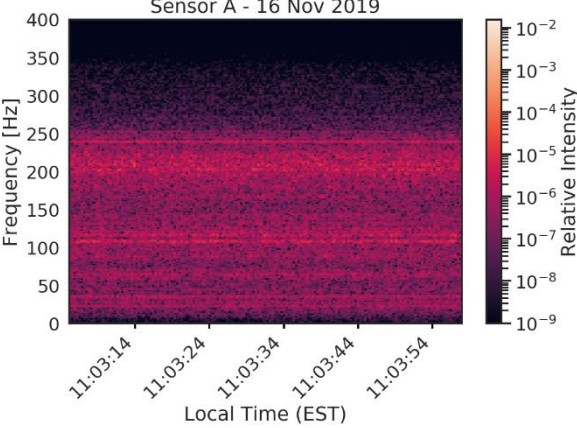

**Figure 3.** Background noise in frequency domain.

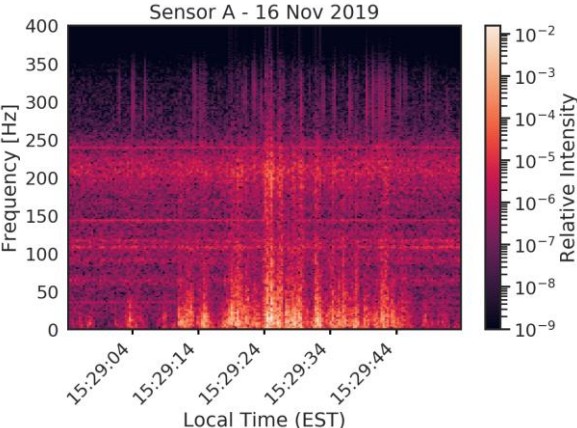

**Figure 4.** Wind noise in frequency domain.

While the nature of the vehicle signature in the frequency domain is relatively easy for a human to recognize, we emphasize that the considerable variability in these signals (cf. Figure 2a,b) and the nonstationary nature of the signature makes traditional audio analyses inappropriate for this detection task. Simple techniques such as loudness detection will produce many false positives from other background sources, while frequency-domain techniques such as marking a set of target frequencies will fail owing to the nonstationary nature. Furthermore, although the vehicle signature was apparent to the human operator, manually scanning through spectrograms associated with multiple days and multiple sensors to locate and identify data associated with events of interest is time intensive. This problem is compounded by the need for a diverse set of negative examples that encompass all possible nonevents (or as many as reasonably feasible). These may include heavy machinery, planes, wind, and people. Typically, these negative examples have lower volume signals (making detecting events more difficult) and more subtle signatures (making attribution more difficult and time consuming).

*2.2. Reference Classifier and Canonical Supervised Learning*

A reference classifier based on a multilayer perceptron was constructed using a tanh activation function with two hidden layers of size 75 and 25, based on the optimized network architecture found by Hite et al. [1]. The scikit-learn machine learning package was used to implement the multilayer perceptron classifier [10]. The multilayer perceptron algorithm assigns "soft" labels to an input feature vector x. Rather than predict a class label directly, the classifier assigns a score of $s(x, \ell) \in [0, 1]$ to each possible label, $\ell$ [11]. This soft label, $s(x, \ell)$, can be interpreted as an indicator of confidence that x belongs to class with label $\ell$ but should not be taken as a probability of class membership without further considerations [12]. A user can then assign a minimum confidence threshold for classification, $\tau$, which can be adjusted to balance sensitivity and the rate of false positives. A trained classifier is used to compute scores and assign a predicted label $\hat{\ell}$ according to Equation (1).

$$\hat{\ell}(x; \tau) = \begin{cases} \text{vehicle} & s(x, \text{vehicle}) \geq \tau \\ \text{background} & \text{otherwise} \end{cases} \tag{1}$$

Training data for the reference classifier consisted of manually labeled examples of vehicle traffic and background from 4 sensor-days collected in late 2019 and early 2020. The collection dates and sensor location associated with these data are shown in Table 1. The vehicle class data are sparser than the background class; training data for the vehicle class were extracted using Equation (1), which resulted in 3805 samples, and an equivalent number of background class samples was drawn. A single day's data from sensor B was reserved for testing and validation. For the test set, 750 examples were drawn from each class.

**Table 1.** Hand-labeled data used to generate reference classifier with supervised learning and for evaluating self-supervised methods.

|  | | Date | Sensor |
| --- | --- | --- | --- |
| Training Data | Friday | 15 November 2019 | A |
| | Friday | 20 December 2019 | B |
| | Thursday | 14 November 2019 | A |
| | Friday | 3 January 2020 | B |
| Testing Data | Tuesday | 17 December 2019 | B |

The manually labeled training data were also used to test our self-supervised methods, described in Section 2.3. Before any work was performed, a reference classifier was trained to distinguish vehicle signatures from background noise using a corpus of hand-labeled data. Classifier performance is measured by the area under the receiver-operating characteristic curve (AUROC), in which the receiver-operating characteristic (ROC) curve is

a parametric plot of the empirical true positive rate versus the false positive rate as a function of threshold $\tau$. This reference classifier (AUROC = 0.9396) was used as a basis for comparison in evaluating our self-supervised learning method.

The confusion matrix for the trained reference classifier is shown in Figure 5, where an operating point of $\tau = 0.47$ is manually selected to meet a true positive rate of 0.7 on the identification of vehicles. Note that compared with the original Hite classifier, for a similar true positive rate (0.69) on the identification of vehicles, this reference classifier offers approximately a 5 percent increase in the true positive rate and a 5 percent decrease in the false positive rate on the identification of the background class.

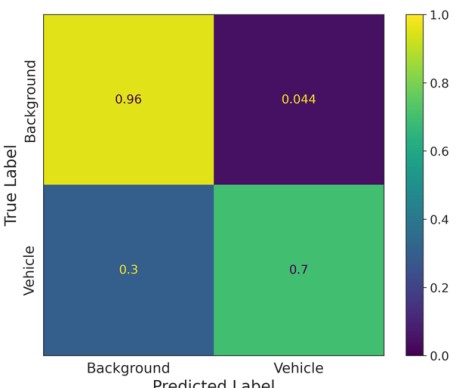

**Figure 5.** Confusion matrix for the trained reference classifier with $\tau = 0.47$ evaluated on the test dataset.

In a field deployment scenario, the classifier is provided with a *stream* of data from the sensor, which is continuously evaluated over a moving window, rather than presented with independent examples. Vehicles at the MINOS testbed are typically audible for 10 s to 60 s, which provides the classifier multiple "attempts" to detect a vehicle. To better understand why a 0.7 true positive rate on vehicle detection is acceptable, we can approximate vehicle detection as a steady-state Bernoulli process. If we use 70% as an estimate of the probability of detecting a vehicle when there is in fact a vehicle present, then after four attempts, the probability of at least one detection exceeds 99%, whereas the probability of at least two detections exceeds 91%. Conversely, using 4.4% as an estimate of the false alarm rate yields a probability of 16% for a single alarm and only 1% for two alarms in the same duration when no vehicle is present. It is difficult to determine whether this approximation is pessimistic or optimistic because successive observations are correlated; the general behavior that is predicted here is consistent with our observations on real data so far but warrants further study.

### 2.3. Semi-/Self-Supervised Learning

Semi-supervised learning refers to using labeled and unlabeled data to perform learning tasks. Labeling data often requires expensive and time-consuming human annotation; therefore, it can be advantageous to use unlabeled data, which often are widely available in many use cases. At the MINOS testbed, data are persistently collected, resulting in months' to years' worth of data, though it can take a trained human operator approximately 2–3 h to annotate a single day's worth of audio data from a single sensor. As described in the previous example of document classification, the use of unlabeled data can help discover new relationships and features that were not represented in the labeled data, resulting in a more powerful model than was previously possible.

The semi-supervised learning method used in this work falls under a collection of inductive wrapper methods, as reviewed by Van Engelen and Hoos [6], though it differs slightly in its final implementation. More specifically, a subset of these wrapper methods are self-learning methods, which consist of a single supervised classifier (the reference classifier) that is iteratively trained on both labeled data and data that have been pseudo-labeled in

previous iterations of the training algorithm. This paper explores several design choices in the iterative self-learning procedure in which the original labeled data are continuously diluted or outright discarded in later generations. Because of these choices, it is unclear whether it is more appropriate to call our method semi-supervised or self-supervised. We adopt the term "self-supervised" to describe our method.

First, we access a single day's worth of unlabeled audio data (Step 1, Figure 6). In the first iteration of the self-learning procedure, we use the reference classifier to assign labels based on a cut point of the 99.9th percentile of all sample scores (Step 2, Figure 6). Vehicle samples are then drawn based on the procedure described in Section 2.2, and an equal number of hand-labeled background class samples are added to these pseudo-labeled vehicle samples to create a balanced training data set (Step 3, Figure 6). Note that the reference classifier's hand-labeled dataset is discarded. Then, the next generation's classifier is trained using the pseudo-labeled vehicle samples and hand-labeled background samples (Step 4, Figure 6) and tested on a holdout hand-labeled dataset consisting of 750 vehicle and background class samples (Step 5, Figure 6). If the new classifier performs worse on the testing set, it, along with its training data is discarded, and the previous classifier is re-used when returning to Step 1 of the self-learning procedure. However, if the new classifier's performance on the testing set improves over the previous one, the new classifier is used when accessing and pseudo-labeling the next day's worth of data. The process is iterated on each day's worth of data until no more unlabeled data remain or the next generation classifier's performance consistently stagnates or degrades relative to the previous generation.

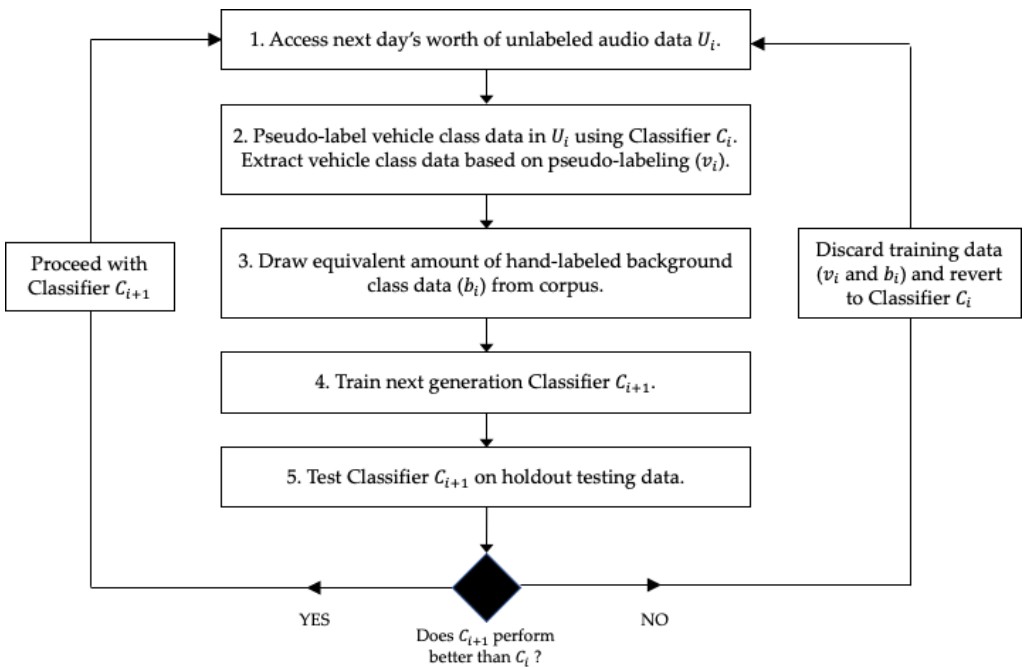

**Figure 6.** Iterative self-learning procedure used in this work.

The conservative cut point of the 99.9th percentile of all sample scores was chosen with the intent of permitting only high-confidence vehicle samples (as determined by the trained classifier) to be added to the training corpus of data, which for each day's worth of audio data resulted in approximately 4–5 min of vehicle data. Early experiments showed that the incremental gains in classifier performance over the generations of the self-supervised learning process were impacted by the choice of cut point, with increments generally increasing with less conservative cut points. However, the ultimate performance seen by a classifier was typically unaffected by this choice and was likely to be rather tied to the quality of data that was fed to the training algorithm. Therefore, we chose a conservative



cut point, with correspondingly smaller incremental gains of classifier performance, given that we had, in effect, infinite unlabeled data and could be reasonably assured of saturating the possible classifier performance.

Several works have examined the application of semi-supervised learning to counteract the issue of drift compensation in electronic nose systems and encounter the issue of data quality in the self-learning process. De Vito et al. offer a criticism of selecting only the high classification confidence unlabeled samples in self-training as insufficient to provide incremental knowledge to correctly identify the separation hyperplane between classes, stating that only limited improvements in performance can be expected [13]. Instead, De Vito et al. suggest relying on different measures of confidence such as pairwise similarity among labeled and unlabeled samples and the use graph-based similarity in the selection of unlabeled samples in a coffee blends classification problem. Liu et al. (2014) take a similar approach and construct weighted geodesic flow kernels to characterize the drift [14]. Liu et al. (2018) compare uncertainty sampling, query-by-committee, and a novel pool-based active learning method for sample selection, and demonstrate that their novel method achieves greater accuracy on two electronic nose drift databases [15]. The optimal selection of unlabeled samples to include in the training process may be the subject of future work.

The self-learning paradigm introduces many design questions, of which we have considered the reuse of pseudo-labeled data in later iterations and selection of the confidence threshold $\tau$ for pseudo-labeling. Unless otherwise stated, all self-learning experiments presented in Section 3 followed the procedure outlined, with the classifier network architecture matching that of the reference classifier. The self-learning procedure offers approximately a 100-fold increase in the amount of data that can be (pseudo-)labeled in the equivalent wall-time for a human operator.

## 3. Results and Discussion

Section 3.1 presents a series of experiments validating the self-learning procedure outlined in Section 2.3, which vary the treatment of pseudo-labeled data between self-learning iterations. We demonstrate that the performance of the self-supervised classifiers gradually improves with each self-learning iteration, with caveats. In Section 3.2, a series of experiments exploring design choices in the self-learning procedure, as well as use-cases for self-learning, are given. The experiments presented in Section 3.1 utilize the 5 sensor-days of hand-labeled data (4 sensor-days of training data, with 1 sensor-day of holdout test data, see Table 1), whereas the experiments presented in Section 3.2 rely on uncharacterized data that were collected at the MINOS testbed. These data are highly representative of what could be expected from a real, operating site where data are persistently collected and where the data-generating distribution may shift over time.

### 3.1. Self-Learning Validation

The self-learning procedure outlined in Section 2.3 is validated in this subsection through a series of experiments. For these experiments, we created a wrapper class for the reference classifier to aid in exploring the limitations of the self-learning procedure, specifically to answer the question: *how good does the seed classifier need to be for self-learning to succeed?* The seed classifier was the first classifier trained on data pseudo-labeled by the reference classifier.

The reference classifier was trained on four hand-labeled sensor-days of vehicle traffic and background data. The wrapper constructed for these validation experiments took the reference classifier and, based on a user-input probability, overrode the predicted class label with the incorrect label if the sample was in the corpus of hand-labeled samples, or with a randomly generated label if it was not. In effect, by increasing the user-input probability, we created increasingly noisy training datasets during the first pseudo-labeling step, resulting in a decrease in the seed classifier's performance.

The wrapper was more effective in pseudo-labeling data from those sensor-days that were hand labeled. Although our hand labels were sparse relative to the total amount of recorded audio, we expected that the reference classifier would have the highest confidence in portions of the data corresponding to the hand-labeled clips (since these were used to train the reference classifier). Then, the wrapper could deliberately mislabel these samples, as opposed to an unlabeled sample for which it would simply generate a randomly assigned label. Because of the increased utility of the wrapper when it was acting on the hand-labeled sensor-days, we chose to discard the original hand-labeled samples that were used to train the reference classifier. When a percentile cut of the scores is used to select new training data, it is likely that the reference classifier will extract the same clips that were hand-labeled, rather than select "new" unseen data which are more likely to aid in training.

Two methods for treating pseudo-labeled data between iterations of the self-learning procedure were examined, which we term *recycle* and *relabel*. In the *recycle* method, pseudo-labeled samples from previous training iterations were recycled into the newest training data. In the *relabel* method, all previous data were newly pseudo-labeled with the latest classifier iteration. Table 2 illustrates the difference in the two methods, where Classifier *i* is the classifier trained on Dataset *i* (i.e., the *i*th sensor-day of hand-labeled data). One might initially presume that the relabel method is superior, as presumably each successive generation of the classifier is more capable than previous versions; however, it is possible that each generation of the classifier will closely follow the latest batch of data and could mislabel old data. Such an effect would be most powerful in the scenario of substantial drift of the data-generating distribution. In this case, retaining old labels generated by previous classifier generations would be more accurate and would increase the diversity in the corpus of training data.

**Table 2.** Origin of vehicle data labels in self-learning experiments.

| | Audio File | Pseudo-Label Source | |
|---|---|---|---|
| | **Date (Sensor)** | **Recycle** | **Relabel** |
| **Dataset 1** | 15 November 2019 (A) | Reference Classifier | Reference Classifier |
| **Dataset 2** | 15 November 2019 (A)<br>20 December 2019 (B) | Reference Classifier<br>Classifier 1 | Classifier 1 |
| **Dataset 3** | 15 November 2019 (A)<br>20 December 2019 (B)<br>14 November 2019 (A) | Reference Classifier<br>Classifier 1<br>Classifier 2 | Classifier 2 |
| **Dataset 4** | 15 November 2019 (A)<br>20 December 2019 (B)<br>14 November 2019 (A)<br>3 January 2020 (B) | Reference Classifier<br>Classifier 1<br>Classifier 2<br>Classifier 3 | Classifier 3 |

### 3.1.1. Recycle Method

Table 3 presents the progression of classifier performance utilizing the recycle self-learning method, accompanied by the total number of vehicle samples drawn with each training iteration (recall that an equivalent number of hand-labeled background samples are drawn and included in each iteration of the updated training data). The effect of varying the probability of overriding the trained reference classifier output on the initial pseudo-labeling iteration is included in Table 3.

**Table 3.** Progression of classifier performance and total number of vehicle samples drawn utilizing the recycle self-supervised learning method and effect of varying probability of overriding the trained classifier output on initial pseudo-labeling iteration.

| | | Probability of Overriding Trained Classifier Output | | | | | | |
|---|---|---|---|---|---|---|---|---|
| | | **0.00** | **0.05** | **0.10** | **0.15** | **0.20** | **0.25** | **0.50** |
| **AUROC** | **Dataset 1** | 0.6676 | 0.6422 | 0.6308 | 0.6002 | 0.5891 | 0.5559 | 0.4912 |
| | **Dataset 2** | 0.7430 | 0.6602 | 0.5940 | 0.5574 | 0.4788 | 0.4844 | 0.3660 |
| | **Dataset 3** | 0.8447 | 0.8005 | 0.7120 | 0.7420 | 0.5961 | 0.6178 | 0.4843 |
| | **Dataset 4** | 0.8983 | 0.8789 | 0.8023 | 0.8185 | 0.7007 | 0.7304 | 0.5579 |
| **Total Samples** | **Dataset 1** | 152 | 456 | 458 | 434 | 450 | 440 | 460 |
| | **Dataset 2** | 590 | 672 | 816 | 880 | 824 | 822 | 898 |
| | **Dataset 3** | 906 | 1018 | 1.174 | 1218 | 1200 | 1194 | 1222 |
| | **Dataset 4** | 1160 | 1368 | 1508 | 1500 | 1472 | 1544 | 1500 |

The performance of the successive classifiers trained in each iteration of the self-learning procedure consistently improved with the addition of newly labeled data. When the reference classifier's performance is not altered (i.e., the probability of the wrapper overriding trained classifier output is 0), this improvement is monotonic, as shown in the progression of ROC curves in Figure 7a. However, when the performance of the reference classifier is dampened (i.e., the probability of the wrapper overriding trained classifier output is 0.1 or greater), the ROC curves associated with each self-learning iteration no longer monotonically improves (see Table 3). This result suggests that the required seed classifier performance, for this self-learning procedure to reliably succeed, lies between AUROC of 0.6308 and 0.6422. However, Classifier 3's performance improves relative to Classifier 2 even when Classifier 2's AUROC is as low as 0.3660, suggesting that the requisite minimum classifier performance is also tied to the quality of new data. Figure 7b shows the progression of ROC curves when the reference classifier wrapper's probability of overriding the trained classifier output is 0.25. In summary, our proposed generative self-learning procedure can improve classifier performance even when the initial seed classifier is substantially inaccurate; but the exact limit on the required seed classifier performance is tied to the quality of the data analyzed in the autonomous training, pseudo-labeling, and retraining process.

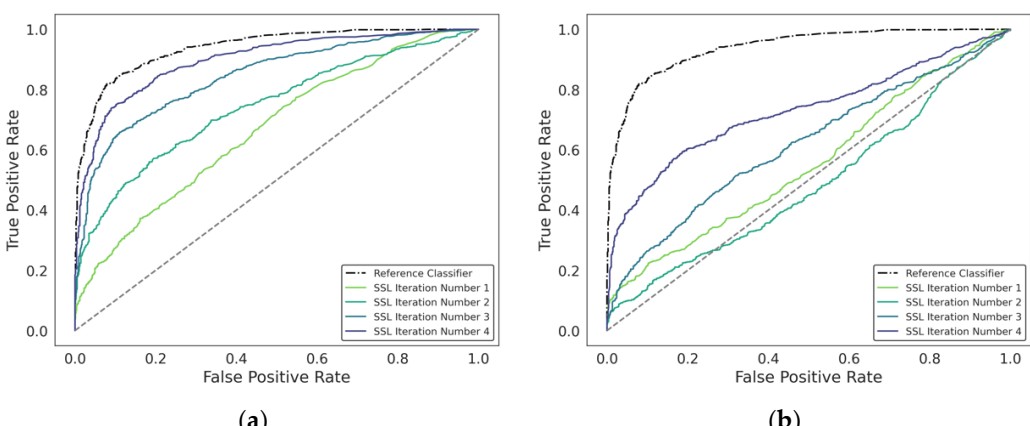

**Figure 7.** Progression of ROC curves utilizing the recycle self-supervised learning method for probability of (**a**) 0.0 and (**b**) 0.25 for overriding trained classifier output on initial pseudo-labeling iteration.

3.1.2. Relabel Method

Table 4 presents the progression of the classifier performance utilizing the relabel self-learning method, accompanied by the total number of vehicle samples drawn with each training iteration (an equivalent number of hand-labeled background samples are

drawn). Table 4 includes the effect of varying the probability of overriding the trained reference classifier output on the initial pseudo-labeling iteration.

**Table 4.** Progression of classifier performance and total number of vehicle samples drawn utilizing the relabel self-supervised learning method and effect of varying probability of overriding the trained classifier output on initial pseudo-labeling iteration.

| | | Probability of Overriding Trained Classifier Output | | | | | | |
|---|---|---|---|---|---|---|---|---|
| | | **0.00** | **0.05** | **0.10** | **0.15** | **0.20** | **0.25** | **0.50** |
| **AUROC** | **Dataset 1** | 0.6676 | 0.6422 | 0.6308 | 0.6002 | 0.5891 | 0.5559 | 0.4912 |
| | **Dataset 2** | 0.5893 | 0.7412 | 0.5489 | 0.7285 | 0.5163 | 0.5533 | 0.3581 |
| | **Dataset 3** | 0.7877 | 0.8267 | 0.8060 | 0.8611 | 0.8096 | 0.7848 | 0.4151 |
| | **Dataset 4** | 0.8815 | 0.8856 | 0.9165 | 0.9038 | 0.8669 | 0.8781 | 0.4257 |
| **Total Samples** | **Dataset 1** | 152 | 456 | 458 | 434 | 450 | 440 | 460 |
| | **Dataset 2** | 544 | 546 | 776 | 892 | 758 | 738 | 812 |
| | **Dataset 3** | 742 | 558 | 620 | 798 | 730 | 700 | 798 |
| | **Dataset 4** | 988 | 858 | 856 | 946 | 962 | 996 | 984 |

An examination of Table 4 shows that the relabel self-learning method yields less consistent improvement in classifier performance across variations in the seed classifier performance. For instance, as shown in the progression of ROC curves in Figure 8a, even when the reference classifier wrapper's probability of overriding the trained classifier output is 0, Classifier 2's performance degrades compared with Classifier 1 but recovers; and the performance of Classifier 4 is approximately equal to its counterpart when the recycle method is used. We speculate that the samples drawn from the recycle method were more diverse, as each successive classifier resulting from the iterative self-learning procedure was tuned to a unique set of features in the acoustic data. Because of this diversity in the training data, the trained classifier was more robust to variations in newly presented data. However, the recycle of training samples from previous iterations in the self-learning procedure may limit the final classifier performance, as there is potential for lower-quality training samples to be included in the training data that are inherited from early iterations. In fact, we did see two instances where the last iteration's trained classifier from the relabel method outperformed the best performing trained classifier from the recycle method.

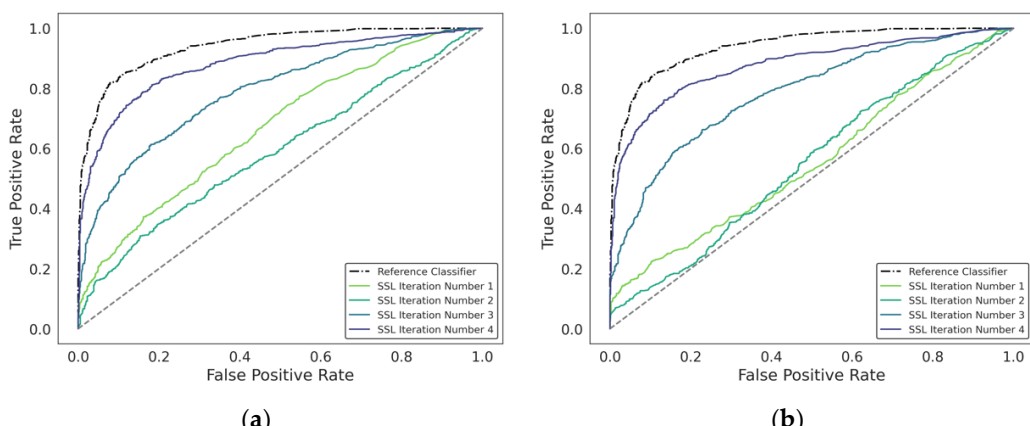

(a)　　　　　　　　　　　　　　　　　　　(b)

**Figure 8.** Progression of ROC curves utilizing the relabel self-supervised learning method for probability of (**a**) 0.0 and (**b**) 0.25 for overriding the trained classifier output on initial pseudo-labeling iteration.

Another feature seen when utilizing the relabel self-learning method was sudden gains (or losses) in classifier performance with each iteration. Figure 8b shows the progression

of ROC curves when the reference classifier wrapper's probability of overriding trained classifier output is 0.25; a large performance gain is observed in the third iteration. Here, we suspect the classifier was able to extract features from clips drawn from the pseudo-labeled vehicle data that were well tuned to the test set.

### 3.2. Extension to Unseen Data

The validation experiments performed in the previous section were conducted on well-characterized sensor-days of data, which were hand-labeled by a human operator. In this section, we use the recycle self-learning method on a month's worth of completely unseen data that were collected at the MINOS testbed in April 2020 (recall that the reference classifier training data consisted of 4 days spanning from November 2019 to early January 2020; see Table 1).

#### 3.2.1. Reference Classifier Training Dataset Retention

The decision to discard the reference classifier training dataset before training successive-generation classifiers on the pseudo-labeled data was influenced by the use of the sensor-days' data listed in Table 1 for the validation experiments (see discussion at the beginning of Section 3.1 on use of the reference classifier wrapper). In this section, we examine the impact of retaining the reference classifier training dataset on subsequent self-learning iterations.

Figure 9 shows the progression of ROC curves when the self-learning procedure is applied to unseen April 2020 data, with a single sensor-day's worth of data ingested per iteration, when the reference classifier training dataset is discarded (Figure 9a, final AUROC of 0.9343) and when it is kept (Figure 9b, final AUROC of 0.9460). The final gain in performance when the reference classifier training dataset is retained is modest, though it still demonstrates that the self-learning procedure offers value over training solely on hand-labeled data.

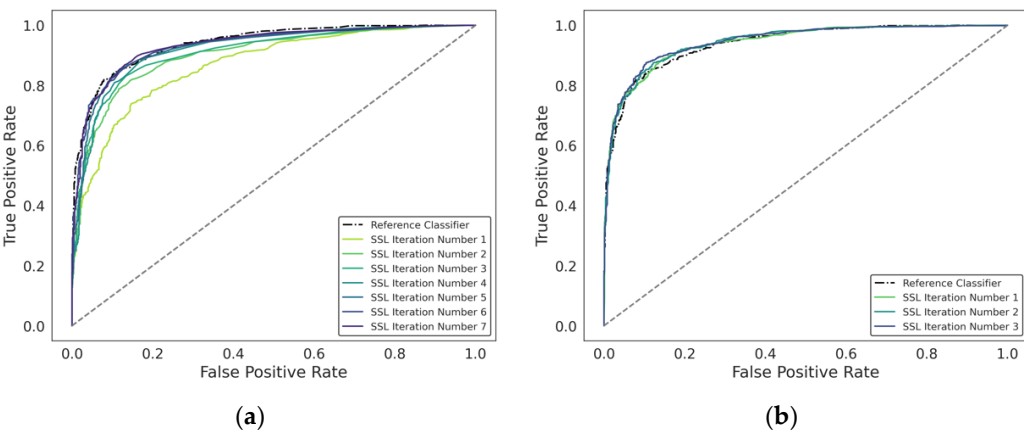

(a)  (b)

**Figure 9.** Progression of ROC curves utilizing the recycle self-supervised learning method when original reference classifier training dataset is (**a**) discarded and (**b**) kept in subsequent training iterations.

#### 3.2.2. Pseudo-Labeled Background

This section explores the effect of using previously trained self-supervised classifiers to pseudo-label background class samples, rather than relying on a holdout pool of hand-labeled background class samples. Figure 10 shows the progression of ROC curves when self-supervised classifiers are used to pseudo-label the background class, resulting in a final classifier with AUROC of 0.9233. Visually, early generations of self-supervised classifiers trained with both pseudo-labeled vehicle and background class samples generally offer an increased false positive rate (the intersection of the "elbow" of the ROC curve tends to fall further right on the *x*-axis). However, as the self-learning procedure continues, this effect is remedied, and the final self-supervised classifier outperforms those presented in Section 3.1.

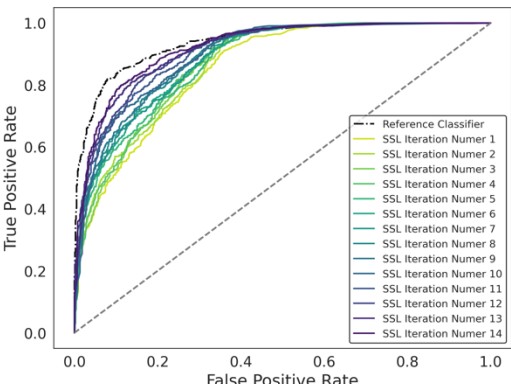

**Figure 10.** Progression of ROC curves utilizing the recycle self-supervised learning method coupled with simultaneous pseudo-labeling of both background and vehicle classes.

### 3.2.3. Sensor Recalibration

Sensor drift is a common issue often caused by environmental changes or contamination, material degradation, or other factors that can manifest as model inaccuracies over time. Therefore, the effect of sensor drift was considered on the performance of the developed iterative self-learning procedure. Figure 11a shows the effect of sensor drift on the performance of the reference classifier that was trained on data from November 2019 to early January 2020, which drops from AUROC = 0.9396 (black line, testing data from December 2019) to AUROC = 0.8525 (red line, testing data from April 2020). We attribute this degradation in classifier performance, as evidenced by the stark movement of the ROC curve, to changes in the underlying data distribution. In terms of the testbed environment, this distribution change(s) was most plausibly associated with changes in season, changes in operational conditions, or changes in the sensors themselves.

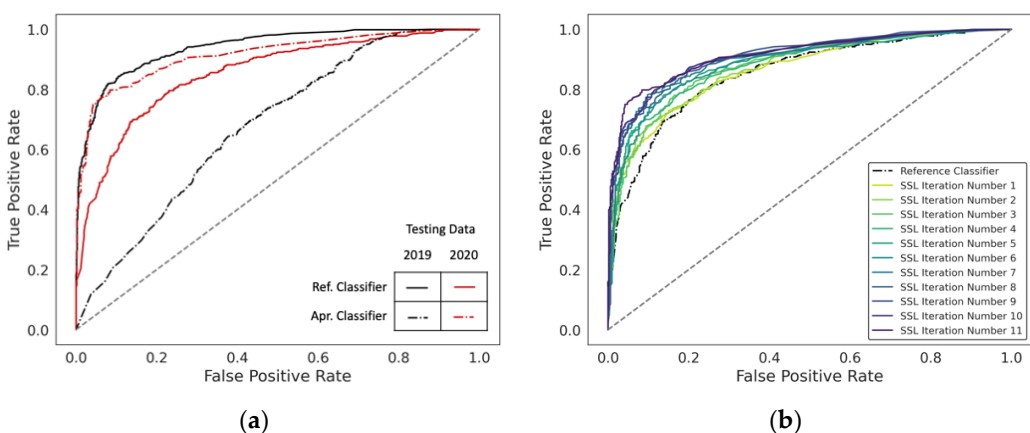

(**a**)  (**b**)

**Figure 11.** (**a**) Performance of the reference classifier on original testing data (dated December 2019) and new testing data (dated April 2020), demonstrating the effects of sensor drift over time. (**b**) Progression of ROC curves associated with self-learning applied for automatic sensor recalibration.

Our autonomous self-learning technique was used to ingest and process data collected in April 2020, closer in time to the April 2020 test data (red line in Figure 8a). Note that the autonomously processed training data collected in April 2020 are separate from the April 2020 data held out for evaluation. The results of testing against these evaluation data after each generation of the self-learning procedure are shown in Figure 11b. With each successive generation, the performance improves (the ROC curve moves up and to the left towards the theoretical optimal step function at false positive = 0, true positive = 1); and after approximately 10 self-learning iterations, the performance of the classifier is comparable to that observed when the original classifier is tested (i.e., the black curve in Figure 11a).

A natural question arises given this result: is the self-learning procedure recalibrating the classifier to account for changes in the data-generating distribution, or is the process expanding the complexity or capacity of the model by virtue of increasing the diversity in the training data? After the classifier was updated using 11 sensor-days of data and associated self-learning iterations, the resulting classifier was retested against the original hold-out testing data collected in December 2020; these results are shown in Figure 11a. We see the performance of the updated classifier is poor when evaluated against the older testing data. From this, we conclude that in this case, the self-learning procedure is functioning as an automatic recalibration procedure; and the updated classifier is not backward compatible. In practice, this behavior is not limiting, as the recalibration procedure is autonomous and not computationally prohibitive and thus may be frequently repeated to consistently keep up with changes in the data-generating distribution over time. We additionally note that this behavior may be different for other scenarios, data collections, or distributions.

Finally, Figure 12 shows the rate of false positives given an assumed fixed rate of 0.7 true positives that are associated with each successive generation of the self-supervised classifiers for automatic sensor recalibration. In the hypothetical Bernoulli process discussed in Section 2.2—when the trained classifier presented in Hite et al. [1] (false positive rate of 9.2%) was compared with our reference classifier (false positive rate of 4.4%)—there was a two-fold reduction in the probability that a single alarm would result after four attempts at vehicle detection when no vehicle was present. This hypothetical Bernoulli process illustrates the impact of small reductions in the false alarm probability, even when there is only a mediocre rate of true positives in vehicle detection.

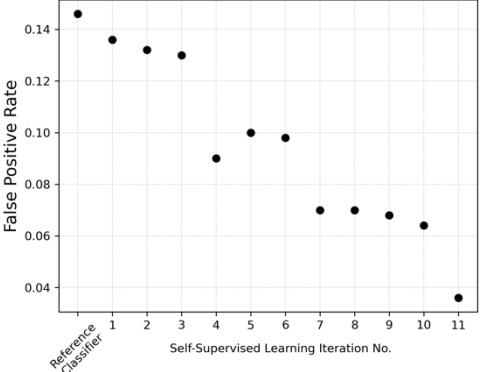

**Figure 12.** Progression of the rate of false positives, at a fixed true positive rate of 0.7, during self-learning procedure applied for automatic sensor recalibration.

## 4. Conclusions

Under the MINOS venture, large volumes of multimodal data have been collected at HFIR and the collocated REDC. However, a fundamental limitation in developing actionable and interpretable analytics using these data is the resource-intensive data identification, curation, and annotation process required to develop appropriate training datasets. Moreover, manually populating the training dataset was determined to be limiting our use case, as this approach was virtually guaranteed to miss some salient phenomena that should be included to make the training data effective (as opposed to many canonical data science problems in which classes and examples are better constrained, defined, or otherwise limited). Thus, our team developed and demonstrated an autonomous method for identifying and annotating training data using an iterative self-supervised procedure.

The approach assumed an analyst can train an initial model using limited available data and use this model to seed a generative approach, in which the current model is applied to unseen data to identify and label interesting examples, which are added to the corpus of training data; a new model is trained with the updated dataset; and the process is repeated. Using a small selection of manually labeled data as an experimental dataset, several initial experiments were performed to assess the efficacy of the method and evaluate

multiple procedural details—including how to select and define useful training examples, identification of background examples, and the effect of the accuracy of the initial classifier used to initialize the iterative procedure. These experiments showed the method to be effective almost entirely independent of the quality of the initial classifier used to seed the process, as new data progressively diluted detection and labeling errors made by the seed classifier. This dilution process was accelerated if the labels for all the examples in the training set were regenerated with each new generation's classifier, resulting in faster improvements to the classifiers' performance, as demonstrated by the ROC curves and associated AUROC values. Moreover, we demonstrated that our method may be used to update a classifier to overcome the effects of data distribution drift stemming from changes in the testbed facilities over months of persistent data collection under the MINOS venture.

Moving forward, we foresee three primary use cases of our self-supervision approach.

**Enabling High-Capacity Models in Limited Data Scenarios:** Many of the recent advances in data science and applications therein have relied on large models (i.e., deep learning) that may typically include millions of tunable parameters. For example, the initial sets of residual nets used to beat the state-of-the-art performance in image recognition used 1.28 million training examples to tune 1.7 million parameters [16]; and transformer models for 1-to-1 language translation or speech separation used 36 million sentences to train 65 million parameters or 108 million audio clips to train 26 million parameters, respectively [17,18]. In many examples, collecting and/or annotating a training dataset is resource prohibitive, which makes the application of such complex models infeasible for these use cases. We propose a multi-stage solution whereby a low-capacity model (e.g., a multilayer perceptron [19]) is trained on a limited training set and used to initialize the iterative self-supervision approach to grow the corpus of training data (wherein each generation of new data is of higher quality by virtue of the improvement in detection performance), until enough labeled training data are available to effectively train the target high-capacity model.

**Automatic Model Recalibration for Long-Term Deployment:** As shown in our experience in the MINOS venture, data may change over the course of a long-term persistent data collection effort. To account for these changes in the data-generating distribution, analytic models must be recalibrated through at least a partial new training routine, which requires new training data. In many scenarios, obtaining these data may be resource intensive (see above) or outright impossible (e.g., if the model was originally trained on data collected in a controlled environment but is deployed in an access-denied area or facility). In these cases, we propose that our self-supervision procedure may be used to periodically develop new training data from the current data-generating distribution and retrain the model. Using this autonomous recalibration process, a model may be persistently deployed long term without a substantial penalty for analyst time.

**Transferring Models without New Training Data:** If a model is initially developed in one data collection effort, new data are typically needed to apply the model in a new scenario, facility, or time. We propose our self-supervision procedure may be used to facilitate this model transfer with substantially reduced analyst or data collection effort. Because our approach has been shown capable of delivering an effective model even when the performance of the initial model used to seed the iterative process is relatively poor (cf. Figures 7 and 8), we believe this transfer approach would be successful even if there are potentially substantial differences between the data-generation distributions of the two scenarios. Note that this statement must be considered with the caveat that the fundamental learning tasks (e.g., classes or data format) must be the same; however, the details of the underlying signatures associated with each class are assumed to be different between the two data collections.

**Author Contributions:** Conceptualization, J.H. and K.D.; methodology, J.H. and B.P.; software, J.H. and B.P.; validation, B.P. and J.H.; formal analysis, B.P.; resources, D.C.; data curation, D.C. and J.H.; writing—original draft preparation, B.P.; writing—review and editing, J.H. and K.D.; visualization, B.P.; supervision, K.D.; project administration, K.D. and J.J.; funding acquisition, J.J. All authors have read and agreed to the published version of the manuscript.

**Funding:** This work was funded by the Office of Defense Nuclear Nonproliferation Research and Development (NA-22), within the US Department of Energy's (DOE) National Nuclear Security Administration. This manuscript has been authored by UT-Battelle, LLC, under contract DE-AC05-00OR22725 with the US DOE. By accepting the article for publication, the US government along with the publisher acknowledges that the US government retains a nonexclusive, paid-up, irrevocable, worldwide license to publish or reproduce the published form of this manuscript, or allow others to do so, for US government purposes. DOE will provide public access to these results of federally sponsored research in accordance with the DOE Public Access Plan (http://energy.gov/downloads/doe-public-access-plan).

**Institutional Review Board Statement:** Not applicable.

**Informed Consent Statement:** Not applicable.

**Data Availability Statement:** Raw infrasound data collected by the MINOS venture at ORNL and used in this study are persistently hosted at https://www.minos.lbl.gov. Access is controlled by the Office of Defense Nuclear Nonproliferation Research and Development (NA-22) within the US DOE's National Nuclear Security Administration.

**Acknowledgments:** The team would like to acknowledge several performers associated with the MINOS venture and the operation of HFIR and REDC, without whom this work would not have been possible. Specifically, we would like to acknowledge Riley Hunley, who was invaluable for explaining and tracking operations at REDC. In addition, the authors would like to acknowledge James Ghawaly for his aid in providing ground truth information to evaluate the efficacy of the team's analyses.

**Conflicts of Interest:** The authors declare no conflict of interest. The funders had no role in the design of the study; in the collection, analyses, or interpretation of data; in the writing of the manuscript, or in the decision to publish the results. This manuscript has been authored by UT-Battelle, LLC, under contract DE-AC05-00OR22725 with the US Department of Energy (DOE). The US government retains and the publisher, by accepting the article for publication, acknowledges that the US government retains a nonexclusive, paid-up, irrevocable, worldwide license to publish or re-produce the published form of this manuscript, or allow others to do so, for US government purposes. DOE will provide public access to these results of federally sponsored research in accordance with the DOE Public Access Plan (http://energy.gov/downloads/doe-public-access-plan).

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
