# Peer review of "Improving an Acoustic Vehicle Detector Using an Iterative Self-Supervision Procedure"

_data, 2022_

Round 1

Reviewer 1 Report

The authors should add a (sub/)section with the description of similar works.

The authors should describe with pseudocode the whole algorithm.

Some kind of cross validation should be used for the experiments and not only one test set as described in Table 1.

A statistical test should be used for the comparison of the examined methods.

The authors should explain why the proposed methodology seems to work well and present some information about the time efficiency of their method.

Reviewer 2 Report

In this article, the authors present an iterative self-supervised learning approach, which improves the ability of vehicle detection with limited ground truth in the data collected from sensors.

The proposed iterative self-learning methodology is well presented in Section 2.3. I am convinced that it is a good approach to learn from non-stationary data streams with limited ground truth.

The authors also provide detailed numerical study in Section 3. Specifically, the authors propose two methods to treat the pseudo-labeled data (recycle vs relabel), and compare the methods with experiments. Then the author use the unseen data to demonstrate the performance of the proposed method.

I am happy to see this paper being published. A few minor suggestions/comments here:

1)    Line 241 - 243 : The content has already been explained from Line 61 to 67, seems duplicate. I also agree that it is more appropriate to call the proposed approach self-learning rather than semi-learning. The authors could refer the readers to the definitions of both terms with some reference.

2)    Line 247: should be section 2.2

Reviewer 3 Report

Section 1 must be improved.

-       Authors should emphasize contribution and novelty, the introduction needs to clarify the motivation, challenges, contribution, objectives, and significance/implication.

-       You should introduce the problem in more detail so that the reader is immediately clear about the purpose of your study. Specify better the essential elements of the problem.

-       You should add more information in the introductory part, you should add other works that have also addressed the problem.

-       The bibliographic part is insufficient (only two works have been cited in the entire introduction.

-       You must properly introduce your work, specify well what were the goals you set yourself and how you approached the problem.

-       The authors have to explain in detail why vehicle tracking must be done through the use of noise emissions

-       At the end of the section, add an outline of the rest of the paper, in this way the reader will be introduced to the content of the following sections.

Section 2 must be improved.

-       Describe in detail the equipment used to make the measurements (acoustic sensors). Extract this data from the datasheet of the instrumentation manufacturer. To make reading the specifications of the instruments more immediate, you can insert them in a table, listing the instruments used and the specific characteristics for each.

-       The authors must describe in detail the acoustic sources present on the site. What other sources can cover the primary source?

-       Subsequently, the authors must acoustically characterize the primary source, i.e. the one to be identified.

-       The presentation section of the methodology needs to be enriched. The authors should present the classification methodology based on machine learning in more detail.

Section 3 must be improved.

-       A description of the hardware and software used for data processing is completely missing. Describe in detail the hardware used:  Extract this data from the datasheet of the hardware manufacturer. To make reading the specifications of the hardware more immediate, you can insert them in a table, listing the instruments used and the specific characteristics for each.

-       Also, you should describe in detail the software platform you used.

-       Also describe the machine learning-based libraries you used.

-       I didn't find a description of the dataset used, how many records did it contain, how many features? How are you divided between training data and test data?

-       I could not find a detailed description of the evaluation metrics you have adopted. How will you measure your model's performance? This section is essential in order to demonstrate the effectiveness of your methodology. Furthermore, only by adopting adequate metrics will it be possible to compare your results with those obtained by other researchers.

-       I did not find a description of the parameters of the methodology

-       At the end of the section there is no adequate discussion on the results obtained. You should summarize the results obtained and compare them with those of other workers. Then you should discuss those results by highlighting the strengths and weaknesses of your methodology.

Section 4 must be improved.

-       Paragraphs are missing where the possible practical applications of the results of this study are reported. What these results can serve the people, it is necessary to insert possible uses of this study that justify their publication.

-       They also lack the possible future goals of this work. Do the authors plan to continue their research on this topic?

Round 2

Reviewer 1 Report

The paper could be accepted in the current form

Reviewer 3 Report

The authors addressed the reviewer's comments with attention and modified the paper with the suggestions provided. The new version of the paper has improved both in the presentation and in the contents